# Comparative Analysis of Differential Cellular Transcriptome and Proteome Regulation by HIV-1 and HIV-2 Pseudovirions in the Early Phase of Infection

**DOI:** 10.3390/ijms25010380

**Published:** 2023-12-27

**Authors:** Tamás Richárd Linkner, Viktor Ambrus, Balázs Kunkli, Zsófia Ilona Szojka, Gergő Kalló, Éva Csősz, Ajneesh Kumar, Miklós Emri, József Tőzsér, Mohamed Mahdi

**Affiliations:** 1Laboratory of Retroviral Biochemistry, Department of Biochemistry and Molecular Biology, Faculty of Medicine, University of Debrecen, 4032 Debrecen, Hungary; linkner.tamas@science.unideb.hu (T.R.L.); ambrus.viktor@med.unideb.hu (V.A.); kunkli.balazs@med.unideb.hu (B.K.); zsofia.szojka@gmail.com (Z.I.S.); 2Doctoral School of Molecular Cell and Immune Biology, University of Debrecen, 4032 Debrecen, Hungary; kumar.ajneesh@med.unideb.hu; 3Division of Medical Microbiology, Department of Laboratory Medicine, Lund University, 22100 Lund, Sweden; 4Proteomics Core Facility, Department of Biochemistry and Molecular Biology, Faculty of Medicine, University of Debrecen, 4032 Debrecen, Hungary; kallo.gergo@med.unideb.hu (G.K.); cseva@med.unideb.hu (É.C.); 5Department of Medical Imaging, Division of Nuclear Medicine and Translational Imaging, Faculty of Medicine, University of Debrecen, 4032 Debrecen, Hungary; emri.miklos@med.unideb.hu

**Keywords:** HIV-1, HIV-2, transcriptomic analysis, proteomic analysis, infection, host response

## Abstract

In spite of the similar structural and genomic organization of human immunodeficiency viruses type 1 and 2 (HIV-1 and HIV-2), striking differences exist between them in terms of replication dynamics and clinical manifestation of infection. Although the pathomechanism of HIV-1 infection is well characterized, relatively few data are available regarding HIV-2 viral replication and its interaction with host–cell proteins during the early phase of infection. We utilized proteo-transcriptomic analyses to determine differential genome expression and proteomic changes induced by transduction with HIV-1/2 pseudovirions during 8, 12 and 26 h time-points in HEK-293T cells. We show that alteration in the cellular milieu was indeed different between the two pseudovirions. The significantly higher number of genes altered by HIV-2 in the first two time-points suggests a more diverse yet subtle effect on the host cell, preparing the infected cell for integration and latency. On the other hand, GO analysis showed that, while HIV-1 induced cellular oxidative stress and had a greater effect on cellular metabolism, HIV-2 mostly affected genes involved in cell adhesion, extracellular matrix organization or cellular differentiation. Proteomics analysis revealed that HIV-2 significantly downregulated the expression of proteins involved in mRNA processing and translation. Meanwhile, HIV-1 influenced the cellular level of translation initiation factors and chaperones. Our study provides insight into the understudied replication cycle of HIV-2 and enriches our knowledge about the use of HIV-based lentiviral vectors in general.

## 1. Introduction

Despite significant advances in anti-retroviral therapy (ARV), the acquired immunodeficiency syndrome (AIDS) caused by the human immunodeficiency viruses 1 and 2 (HIV-1 and 2) continues to pose a global health challenge. Both viruses are prominent members of the *lentivirinae* class of the *retroviridae* family. While HIV-1 is responsible for the majority of infections, HIV-2 remains mostly confined to countries in West Africa, although recent studies have shown that the virus is gaining a foothold in the United States, India and Europe [1]. Even though both viruses share the same mode of transmission, and can both result in AIDS, the dynamics of infection and clinical progression are strikingly different. Infection with HIV-1 is known to be more pathogenic, with faster progression to AIDS and higher plasma viral load. On the other hand, infection with HIV-2 is characterized by lower transmissibility, slower decline of CD4 positive T cells and overall, a higher chance for non-progression to AIDS [2,3]. Both viruses share a similar genomic architecture and protein composition. The genome of HIV consists of two copies of positive, single-stranded RNA coding for the main structural capsid and matrix proteins and viral enzymes. Moreover, the genome codes for the regulatory proteins, which are required for efficient viral replication, in addition to auxiliary proteins. The HIV-2 genome does not contain the viral protein u (Vpu) gene but instead contains viral protein x (Vpx), which is also found in the simian immunodeficiency virus of sooty mangabey (SIV/sm) lineages [4,5,6].

The HIV life cycle, like that of all retroviruses, can be divided into an early and late phase. The early phase begins with the binding and entry of the viral particle to a target cell. Approximately 1–2 h post-entry, reverse transcription begins inside the viral core [7]. The pre-integration complex (PIC) enters the nucleus through a complex interplay between viral and host proteins [8]. The ability of HIV to enter the nucleus of non-dividing cells via its accessory proteins makes the virus a perfect base for gene delivery vector systems. It is proven that HIV-based vectors can effortlessly deliver genes into several cell types, including hematopoietic stem cells [9]. Several host proteins associated with the PIC aid the viral genome during integration. LEDGF/p75 is a transcriptional regulator, which is shown to tether the viral genome to the transcriptionally active chromatin sites [10,11].

The end of proviral genome integration marks the transition from the early to late phase in the life cycle of HIV, during which transcription ensues, and new viral particles are assembled [12]. For the transcription of viral RNA, Tat binds to the trans-acting response element (TAR) region of the viral genome and recruits cellular factors, such as protein-positive transcription elongation factor b (P-TEFb), comprising cyclin-dependent kinase 9 (CDK9) and cyclin T1 (CCNT1), allowing for the synthesis of full-length viral RNA and different mRNA products [13].

Multiple studies have focused on the proteomic and transcriptomic changes of cells from latent HIV infection or cells from non-progressing patients. It was previously shown that HIV can be controlled by the infected cells through transcription and cell cycle regulators, such as TGF-β signaling [14,15].

Also, in latently infected CD4+ T-cells, genes related to apoptosis, cell cycle and human leukocyte antigen class II (HLA-II) were upregulated [16]. Moreover, the upregulation of an inflammatory state is also observed in latently infected immune cells [17].

We have previously carried out a comparative analysis between HIV-1 and HIV-2 in the immediate early phase of infection; however, to the best of our knowledge, we are not aware of similar studies on the entire early phase. Previous works focusing on HIV-1 viral replication revealed that proteins related to translation, transcription and DNA condensation were differentially expressed in the early phase [18]. Moreover, the most abundant proteomic changes were detected at 24 h post-infection [19]. Also, differential regulation of metabolic, apoptotic and mitochondrial proteins was observed in HIV-1-infected T cells [20].

Many studies have characterized the impact of HIV-1 infection on target cells; in contrast, data covering virus–host interactions in the case of HIV-2 are lacking. We previously showed significant differences in the virus–host interaction between the two viruses as early as 2 h after transduction in HEK-293T cells.

In this study, we focused our efforts on the analysis of later time-points at 8, 12 and 26 h post-transduction in the same cell line.

## 2. Results

### 2.1. Transcriptomics

#### 2.1.1. Analysis of Transcriptomic Changes at Different Time-Points Following HIV-1 and -2 Transduction

Compared to the mock transduced cells, HIV-1 managed to significantly alter the expression of 158 genes eight hours post-transduction. Of the significantly altered transcripts, 103 coded for proteins, 30 were non-coding RNA and 5 coded for mitochondrial RNA products. Furthermore, 16 pseudogenes and 4 uncategorized gene products were also detected. At 12 h post-transduction, we detected 114 genes, which were regulated differentially by HIV-1. Out of these 114 genes, 70 were protein coding, 5 were products of mitochondrial RNA, 33 coded for non-coding RNAs, 2 for pseudogenes and 4 were the products of unknown genes. The most prominent alteration in the transcriptome of HIV-1 infected cells was detected at the 26 h time-point, wherein a total of 380 gene products were altered, compared to the mock control. Out of the 380 genes, 290 coded for proteins, 70 for non-coding RNA, 15 for pseudogenes and 5 were unknown gene products.

On the other hand, at the 8 h time-point, HIV-2 managed to alter the expression of 283 genes compared with the mock-transduced cells, of which 142 coded for proteins, 58 for non-coding RNA, 3 for mitochondrial RNA and 77 for pseudogenes, and 3 were the products of unknown genes. At the 12 h time-point, the expression of 299 genes were significantly altered by HIV-2, out of which 153 were protein coding, 50 were non-coding RNA, 1 coding for mitochondrial RNA and 93 were products of pseudogenes, while 2 were coding for unknown genes. 26 h post-transduction, 182 genes were differentially expressed, of which 70 were protein coding, 29 were non-coding RNA transcripts, 4 coding for mitochondrial RNA, 78 were products of pseudogenes, and 1 was an unknown gene product. The altered differential gene expression pattern of the transduced cells at the different time-points is presented in Figure 1 and Figure 2, and the full list of the detected significant transcripts can be found in Appendix A.

#### 2.1.2. Differentially Induced Genes by HIV-1

The top 10 DEGs that were protein-coding from HIV-1-transduced HEK cells at different time-points were selected and presented in Figure 3.

At the 8 h time-point, the 10 most upregulated genes by HIV-1 were the leucine-rich repeats and IQ motif containing 1 (LRRIQ1), centromere protein E (CENPE), A-kinase anchoring protein 9 (AKAP9), ankyrin repeat domain 12 (ANKRD12), coiled-coil domain containing 88A (CCDC88A), biorientation of chromosomes in cell division 1 like 1 (BOD1L1), GRIP and coiled domain containing 2 (GCC2), ankyrin repeat domain 26 (ANKRD26), structural maintenance of chromosomes 4 (SMC4) and dopamine receptor D4 (DRD4). The most downregulated protein-coding genes eight hours post-transduction were the WD repeat domain 38 (WDR38), fos proto-oncogene AP-1 transcription factor subunit (FOS), bolA family member 2B (BOLA2B), cyclin-dependent kinase inhibitor 1A (CDKN1A), inhibitor of DNA binding 3, HLH protein (ID3), H4 clustered histone 3 (H4C3), collagen type XI alpha 1 chain (COL11A1), epithelial membrane protein 3 (EMP3), RELB proto oncogene NF-kB subunit (RELB) and pleckstrin homology-like domain family A member 3 (PHLDA3).

At the 12 h time-point, the top 10 upregulated protein-coding genes induced by HIV-1 in decreasing order of magnitude were the secretogranin III (SCG3), forkhead box D4 (FOXD4), neuregulin 4 (NRG4), leucine-rich repeat and coiled-coil and coiled centrosomal protein 1 (LRRCC1), nucleosome assembly protein 1 like 2 (NAP1L2), BRCA2 DNA repair associated (BRCA2), nibrin (NBN), taste 2 receptor member 20 (TAS2R20), cytochrome c oxidase assembly factor COX20 (COX20) and transmembrane protein 145 (TMEM145). Meanwhile, at the same time-point, H4 clustered histone 3 (H4C3), collagen type XI alpha 1 chain (COL11A1), transmembrane protein 132E (TMEM132E), calcium/calmodulin-dependent protein kinase ID (CAMK1D), SH2 domain containing 3C (SH2D3C), tetraspanin 11 (TSPAN11), programmed cell death 11 (PDCD11), NOP9 nucleolar protein (NOP9), early growth response 1 (EGR1) and kinesin family member 1A (KIF1A) were downregulated.

Then, 26 h post-transduction, the levels of heme oxygenase 1 (HMOX1), oxidative stress-induced growth inhibitor 1 (OSGIN1), VGF nerve growth factor inducible (VGF), NAD(P)H quinone dehydrogenase 1 (NQO1), nucleosome assembly protein 1 like 2 (NAP1L2), heat shock protein family A (Hsp70) member 1A (HSPA1A), dehydrogenase/reductase 2 (DHRS2), ETS variant transcription factor 4 (ETV4), metallotioenin 2A (MT2A) and glutamate-cysteine ligase modifier subunit (GCLM) were elevated. Meanwhile, levels of H4-clustered histone 3 (H4C3), kelch domain containing 7B (KLHDC7B), inositol polyphosphate-5-phosphatase D (INPP5D), glutamate ionotropic receptor kainate type subunit 3 (GRIK3), transmembrane protein 132E (TMEM132E), ATP binding cassette subfamily G member 1 (ABCG1), AT-hook transcription factor (AKNA), dehydrogenase/reductase 3 (DHRS3), hes family bHLH transcription factor 5 (HES5) and ABI family member 3 binding protein (ABI3BP) were found to be downregulated by HIV-1. A full list of the detected transcripts can be found in Appendix A. Transcripts showing consistent level changes across time-points are highlighted in Appendix A.

#### 2.1.3. Differentially Induced Genes by HIV-2

We selected and presented the top 10 up- and downregulated protein-coding genes at 8, 12 and 26 h following transduction with HIV-2 in Figure 4.

Eight hours post-transduction, HIV-2 managed to upregulate the expression of collagen type I alpha 2 chain (COL1A2), keratin 5 (KRT5), serpin family E member 1 (SERPINE1), keratin 14 (KRT14), S100 calcium-binding protein A6 (S100A6), keratin 7 (KRT7), decorin (DCN), thrombospondin 1 (THBS1), ETS variant transcription factor 5 (ETV5) and collagen type VI alpha 3 chain (COL6A3). At the same time-point, H4 clustered histone 3 (H4C3), inhibitor of DNA binding 3, HLH protein (ID3), ectodysplasin A2 receptor (EDA2R), AHNAK nucleoprotein (AHNAK), transmembrane protein 132E (TMEM132E), inhibitor of DNA binding 1, HLH protein (ID1), AHNAK nucleoprotein 2 (AHNAK2), paralemmin 3 (PALM3), hes family bHLH transcription factor 5 (HES5) and ArfGAP with dual PH domains 1 (ADAP1) were downregulated.

At the 12 h time-point, the most upregulated protein-coding genes were the collagen type I alpha 2 chain (COL1A2), keratin 7 (KRT7), decorin (DCN), alanyl aminopeptidase, membrane (ANPEP), dermatopontin (DPT), integrin subunit beta like 1 (ITGBL1), keratin 5 (KRT5), transforming growth factor beta induced (TGFBI), S100 calcium-binding protein A6 (S100A6) and fatty acid binding protein 4 (FABP4). Meanwhile, the expression level of H4 clustered histone 3 (H4C3), ATP binding cassette subfamily G member 1 (ABCG1), inhibitor of DNA binding 3, HLH protein (ID3), ankyrin repeat domain 36C (ANKRD36C), transmembrane protein 132E (TMEM132E), tetraspanin 11 (TSPAN11), ankyrin repeat domain 18A (ANKRD18A), regulating synaptic membrane exocytosis 3 (RIMS3), pleckstrin homology like domain family A member 3 (PHLDA3) and calcium/calmodulin-dependent protein kinase ID (CAMK1D) were suppressed by HIV-2 compared with the mock control.

At the 26 h time-point, we detected attenuated expression of collagen type I alpha 2 chain (COL1A2), keratin 7 (KRT7), keratin 5 (KRT5), keratin 14 (KRT14), decorin (DCN), S100 calcium-binding protein A6 (S100A6), serpin family E member 1 (SERPINE1), S100 calcium-binding protein A2 (S100A2), SLX1 homolog B, structure-specific endonuclease subunit (SLX1B) and thrombospondin 1 (THBS1) were upregulated by HIV-2, while H4 clustered histone 3 (H4C3), early growth response 1 (EGR1), kelch domain containing 7B (KLHDC7B), RELB proto-oncogene NF-κB subunit (RELB), ATP binding cassette subfamily G member 1 (ABCG1), transmembrane protein 132E (TMEM132E), SHC adaptor protein 2 (SHC2), dehydrogenase/reductase 3 (DHRS3), tetraspanin 11 (TSPAN11) and AT-hook transcription factor (AKNA) were found to be downregulated. A full list of the detected transcripts can be found in Appendix A, and a detailed list of transcripts showing consistent tendency across time points is provided in Appendix A.

#### 2.1.4. Gene Ontology Analysis of the Significantly Altered Transcripts across the Different Time-Points

To further enrich the data of the detected transcripts, gene ontology (GO) analysis was performed with the most significant, differentially expressed (*p* < 0.05, log_2_FC > 1) protein-coding genes at the 8, 12 and 26 h time-points from both HIV-1 and HIV-2 transduced cells, respectively, compared to mock-transduced cells. Analysis revealed further differences between HIV-1 and HIV-2 transduced cells compared to the mock control.

At 8 h, HIV-1 managed to regulate the expression of genes involved in the cell cycle, positive regulation of metabolic processes and positive regulation of the RNA metabolic process. At the same time-point, HIV-2 regulated the cellular level of genes, which are part of cell adhesion, extracellular matrix organization and tissue development (Figure 5). At the 12 h time–point, there were no significantly affected GO terms in the case of HIV-1 transduced cells. However, HIV-2 managed to alter the expression of genes related to cell differentiation, cell adhesion and cell migration (Figure 6).

Twenty-six hours post-transduction, HIV-1 altered the regulation of genes involved in response to oxidative stress, response to toxic substances and homeostatic processes. Meanwhile, HIV-2 affected genes that are part of the negative regulation of cell motility, negative regulation of cell migration and peptide cross-linking (Figure 7). A full list of the detected GO terms can be found in Appendix A.

### 2.2. Proteomics Analysis

#### 2.2.1. Proteomic Changes Induced by Transduction with HIV-1 and HIV-2

At all time-points, altogether, 1058 unique proteins were detected by MS. 871 proteins at 8 h, 817 at 12 h and 810 at 26 h. Analysis of proteomic data did not show any significantly altered proteins at 8 h post-transduction. At 12 h, there were 17 differentially regulated proteins, out of which 5 were regulated by HIV-1, 4 by HIV-2 and 8 by both viruses, compared with the mock control. Out of the HIV-1 differentially altered proteins, we detected the synaptotagmin binding cytoplasmic RNA interacting protein (SYNCRIP), valosin-containing protein (VCP) and ATP synthase F1 subunit beta (ATP5F1B), among others. HIV-2 significantly altered the expression of proline-rich coiled-coil 2A (PRRC2A), valyl-tRNA synthetase 1 (VARS1), phosphoribosylformylglycinamidine synthase (PFAS) and non-SMC condensin I complex subunit H (NCAPH). Both viruses induced changes in the expression of deoxyuridine triphosphatase (DUT), heat shock protein family A (Hsp70) member 1A (HSPA1A), nucleophosmin 1 (NPM1), arginyl-tRNA synthetase 1 (RARS1) and Zyxin (ZYX).

At 26 h, we detected 117 differentially regulated proteins from HIV-transduced cells. Out of the 117 proteins, 25 exhibited differential regulation by HIV-1, 39 by HIV-2 and 48 by both, in comparison to the mock control; additionally, 5 proteins were exclusively regulated in the HIV-2 vs. HIV-1 comparison.

Among proteins differentially regulated by HIV-1 were the ATPase family AAA domain containing 3A (ATAD3A), HIV-1 Tat-specific factor 1 (HTASF1), protein disulfide isomerase family A member 3 (PDIA3), DEAD-box helicase 3 X-linked (DDX3X) and lysyl-tRNA synthetase 1 (KARS1), proteasome 20S subunit alpha 2 (PSMA2) and insulin-like growth factor 2 receptor (IGF2R). On the other hand, HIV-2 differentially regulated the expression of serine and arginine-rich splicing factor 1 (SRSF1), eukaryotic translation elongation factor 1 alpha 1 (EEF1A1) and nucleolin (NCL), among others. Both HIV-1 and 2 altered the expression profile of serine and arginine-rich splicing factor 2 (SRSF2), heterogeneous nuclear ribonucleoprotein K (HNRNPK), heterogeneous nuclear ribonucleoprotein A2/B1 (HNRNPA2B1), eukaryotic translation initiation factor 2B subunit delta (EIF2B4) and dynein cytoplasmic 1 light intermediate chain 1 (DYNC1LI1). In the HIV-2 vs. HIV-1 comparison, altogether, 17 proteins had their regulation altered. Of utmost importance were the RAN binding protein 1 (RANBP1), ribosomal protein L23a (RPL23A), heat shock protein family A members and transferrin receptor (TFRC).

The altered differential protein expression pattern of the transduced cells at the 12 and 26 h time-points is presented in Figure 8 and Figure 9. Proteins that were differentially regulated in all of the three observed time-points were selected and presented in Figure 10, and the full list of the detected proteins can be found in Appendix A.

#### 2.2.2. Gene Ontology Analysis of the Significantly Altered Proteins

GO term enrichment analysis of the significantly altered proteins was carried out for HIV-1 and HIV-2 transduced samples at 12 and 26 h post-transduction (Figure 11). At 12 h, compared with the mock-transduced cells, HIV-1 altered the expression of proteins involved in mRNA splicing via spliceosome, ATP metabolic processes and ubiquitin protein ligase binding. In contrast, HIV-2 strongly affected members of the negative regulation of macromolecule biosynthetic process, regulation of cellular response to stress and regulation of localization. At 26 h, HIV-1 managed to alter the expression of proteins involved in response to virus, viral genome replication and RNA polymerase II transcription regulator complex. On the other hand, HIV-2 altered proteins are part of the viral genome replication, regulation of viral genome replication, and spliceosomal snRNP complex. The full list of the detected GO terms can be found in Appendix A.

To further enrich the relations between the DEPs and the GO terms, we selected six GO terms from HIV-1 and HIV-2-transduced samples at 12 and 26 h. The relationship between the DEPs and selected GO terms, as well as the logFC of the DEPs, are shown in Figure 12.

## 3. Discussion

Lentivirus-based vectors are widely utilized in research and clinical settings. For safe application, it is important to delineate the changes induced by these vectors in the host cell. HIV-1 and HIV-2 share many similarities, including routes of transmission and clinical outcomes of infection. However, there are striking differences between the two viruses in terms of replication dynamics and clinical course of infection. In comparison to HIV-1, HIV-2 is characterized by an acute “surge” in viral production right after infection; thereafter, a prolonged latency stage ensues, characterized by lower viral replication and a slower rate of disease progression [1,21,22]. Factors involved in the replication dynamics and prolonged latency of HIV-2 remain mostly unknown. While there are adequate data about the proteo-transcriptomic changes induced by HIV-1, data regarding HIV-2 are lacking.

Given the wide divergence in receptor utilization between the two virions, we used VSV-G for pseudotyping in this study, which was shown to utilize the low-density lipoprotein (LDL) receptor as a target [23]. This was carried out in order to bypass the receptor-induced proteo-transcriptomic changes, facilitating the transduction of a wide range of cells. HEK-293T cells, on the other hand, lack restriction factors and antiviral immune response and express the large T antigen, making them widely utilized for lentiviral production, allowing for effective transfection efficiency and, thereafter, robust vector production.

We utilized RNAseq and proteomic analysis to determine changes in the cellular transcriptome and proteome in the early- and early-–late phases of the HIV life cycle, focusing in this study on 8, 12 and 26 h time-points post-transduction. HEK-293T cells transduced with HIV-1 showed a dynamic change in the number of DEGs, changing from 158 at 8 h to 114 at 12 h and, finally, to 380 at 26 h post-transduction. Compared with HIV-1, at the 8 h time-point, HIV-2 differentially regulated the expression of 283 genes, significantly higher than HIV-1 at the same time-point. At 12 h, HIV-2 managed to alter the expression of 299 genes, and the number dropped to 182 at 26 h. The higher number of genes altered by HIV-2 in the first two time-points suggests a more diverse effect on the host cell, followed by a stagnant state at 26 h, compared with HIV-1, which steadily altered the cellular transcriptome (Figure 1).

At 8 h, amongst the most upregulated protein-coding genes by HIV-1 was the A-kinase anchor protein 9 (AKAP9). The AKAP family of proteins has been described as contributors to the cAMP-mediated signaling. Indeed, different health conditions, such as chronic heart failure, cancers and immune deficiencies, were associated with alterations in the expression of AKAP [24]. Dysregulation of the cAMP levels has been associated with T-cell dysfunction, which is beneficial to HIV, as it aids the virus in evading immune response [25]. Moreover, RelB was downregulated by HIV-1 at the 8 h time-point, and it was shown that viral components, such as HIV-1 Vpr, bovine foamy virus (BFV) transactivator (Btas) and human T cell leukemia virus (HTLV1) Tax1 can interact with RelB to facilitate virus replication [26]. Additionally, silencing of RelB can induce a stop at the G1 phase in which HIV gene expression is promoted. Interestingly, the HIV-1 Tat protein can also induce G1 arrest and is capable of interacting with RelB to promote viral gene expression [26,27,28]. Moreover, HIV-1 generated G1-like state in infected macrophages can promote HIV-1 replication by bypassing SAMHD1-mediated restriction [29]. Taken together, it is conceivable that HIV-1 Tat protein may induce G1 arrest through the downregulation of RelB expression while using the available RelB protein to promote bypassing cellular restrictions in the early phase and promote gene expression in the late phase of infection.

Additionally, expression of genes coding for the group and coiled-coil domain containing 2 (GCC2) were upregulated 8 h after transduction with HIV-1. GCC2 was implicated in the Nef-mediated downregulation of MHCI, thereby protecting the infected cell from CD8 + T-cell-mediated killing [30].

The cyclin-dependent kinase inhibitor 1 A (CDKN1A) was downregulated after eight hours following transduction with HIV-1. It was previously shown that CDKN1A was able to inhibit the integration of HIV-1 into hematopoietic stem cells by forming complexes with the viral integrase [31]. It was also revealed that HIV-1 Vpr can activate the cellular expression of CDKNA1A and initiate cell cycle arrest in macrophages and T cells, thus aiding the integration of the PIC [32,33]. All in all, the regulation of CDKN1As can positively influence the life cycle of HIV, especially the reverse transcription and integration events, although this regulation might be tied to other cellular factors and could vary according to the type of the infected cell.

At 12 h post-transduction, HIV-1 enhanced the expression of COX20, a member of the Cytochrome C oxidase complex and an essential part of the mitochondrial respiratory chain complex IV [34]. The p2 peptide, which is released during HIV-1 uncoating in the early phase of the viral life cycle, can activate cytochrome C oxidase and increase ATP production to enhance HIV-1 reverse transcription and nuclear import [35]. Later on, during viral replication, the HIV-1 Tat protein inhibits the activity of cytochrome C oxidase, which leads to Tat-mediated apoptosis [36].

Also, calcium/calmodulin-dependent protein kinase 1D (CAMK1D) was downregulated by HIV-1. The activity of CAMK1D was found to be negatively regulated by HIV-1 infection, which was in correlation with our results, wherein we observed decreased mRNA levels [37]. It was also shown that siRNA knockdown of CAMK1D can inhibit HIV-1 infection [38].

As genome integration begins roughly 15 h post-infection, upregulation of protein-coding genes involved in the maintenance of DNA structure is to be expected near that time-point [39]. Indeed, our data showed that genes of proteins involved in the DNA break repair, such as BRCA2 and NBN, were augmented 12 h post-transduction [40,41].

Finally, after 26 h of transduction, several protein-coding genes involved in the oxidative stress response were upregulated by HIV-1. HIV infection generates reactive oxygen species, resulting in the deregulation of oxidative stress pathways and induction of mitochondrial dysfunction [42]. Genes coding for hem-oxygenase 1 (HMOX1), oxidative stress-induced growth inhibitor 1 (OSGIN1) and NAD(P)H quinone dehydrogenase 1 (NQO1) were upregulated by HIV-1, surmounting active defense against the generated oxidative stress.

In our previous publication, we described that changes generated by transduction with HIV-2 pseudovirions were significantly different from those generated by HIV-1 [43]. Indeed, this pattern was also noticed in later time-points as well.

At the eight-hour time-point, HIV-2 managed to differentially alter the inhibitor of DNA binding protein 1, HLH protein (ID1) and 3 (ID3), compared with the mock control. ID proteins are a class of helix-loop-helix transcription factors that are involved in cell proliferation, cell cycle regulation and cell migration [44]. Additionally, decorin (DCN)—a protein involved in intracellular communication and cell proliferation—was continuously upregulated by HIV-2 in all of our observed time-points. Overexpression of DCN was shown to induce apoptosis through caspase-3 and cause cell cycle arrest at the G0–G1 phase by upregulating the expression of p21 [45,46].

HIV-2 also managed to upregulate the expression of S100A6, a calcium-binding protein involved in cell proliferation, cytoskeletal functions and cellular response to stress. Elevated S100A6 levels were shown to induce cell proliferation and migration in cancer cells [47,48]. The differential regulation of genes involved in cell proliferation is important for the integration of viral DNA. Moreover, the control of cell proliferation is important for the expansion of the viral reservoir and is a key part of HIV’s effect on the cell [49,50].

Amongst the differentially expressed genes by HIV-2 at 12 h post-transduction were genes—products of which are involved in lipid homeostasis. Patients tend to develop lipid homeostatic abnormalities called lipodystrophy due to HIV infection and subsequent therapy [51,52]. mRNA of fatty acid binding protein 4 (FABP4)—which is a strong biomarker of HIV, generated metabolic syndrome and lipodystrophy—was significantly upregulated by HIV-2 but not by HIV-1 even at later time-points [53]. An important finding was the downregulation of ATP binding cassette subfamily G member 1 (ABCG1), a key transporter in cholesterol efflux. It was revealed in a study that HIV-1 Nef is capable of inhibiting the efflux of cholesterol through downregulation of the ATP binding cassette subfamily a member 1 (ABCA1) transporter but has little effect on the ABCG1. Impairment in the cholesterol homeostasis of the transduced cells is an important part of HIV replication, as sufficient cholesterol is required for the production of nascent viral particles; additionally, inhibition of the cholesterol efflux promotes inflammatory macrophage differentiation by inducing TLR signalization [54,55]. As HIV-2 did not alter ABCA1, even at later time-points, and only altered ABCG1 at the 12 and 26 h, it appears that HIV-2 does not affect the cellular cholesterol level. Moreover, it is possible that the induction of inflammation at an early stage of HIV infection through dysregulation of the cellular cholesterol level in the early phase of the viral life cycle can initiate an early immune response against HIV-2-infected cells. Since HIV-1 was able to alter the expression of ABCA1, ATP binding cassette subfamily a member 2 (ABCA2), and even ABCG1 at the 26 h time-point, we can speculate that HIV-1 has a more noticeable effect over the cellular cholesterol transport compared with HIV-2 (Appendix A).

In all of our observed time-points, HIV-2 resulted in the upregulation of transcripts coding for proteins associated with the extracellular and intracellular matrix, such as keratins (KRT4, 5, 7) and collagens (COL1A2, COL6A3). Meanwhile, HIV-1 only managed to alter the expression of collagens (COL11A1) but not keratins. There is no available information on keratins playing any role in the life cycle of the virus; however, HIV-1 was shown to induce the remodeling of the extracellular matrix through the impairment of metalloproteinases [56]. Dynamic changes in the extracellular matrix are necessary for cellular processes, such as cell proliferation, tissue repair and the procession of biomolecules, including chemokines, growth factors and cytokines. Indeed, for a successful immune response against pathogens, remodeling of the extracellular matrix is needed, and alteration in the composition of the matrix can limit the ability of the immune cells to effectively combat infections [57]. Moreover, remodeling of the intracellular matrix is required for the efficient transport and assembly of viral particles [58,59]. Both HIV-1 and 2 managed to alter the expression of EGR1, a transcription factor involved in Tat-dependent HIV gene expression. EGR1 is important in the therapeutic strategy “kick and kill”, wherein latently infected cells are treated with latency reversal agents in order to reactivate latent HIV reservoirs. EGR1 was upregulated following treatment with various agents, and it was shown to directly interact with the HIV-1 promoter to induce viral transcription [60,61].

An interesting finding was the regulation of different pseudogenes by both HIV-1 and HIV-2 in all of the observed time-points. Previously, pseudogenes were described as functionless DNA created by mutations, frame-shifts and gene duplication events. However, in the past few years, their role in cellular homeostasis has been reimagined as it was discovered that pseudogenes can regulate the expression of their parent genes [62]. Moreover, evidence suggests that self-derived mRNAs play an important role in the regulation of immune response against viruses and tumors. For example, 15S and 5S ribosomal RNA-derived pseudogenes can increase the expression of proinflammatory cytokines [63]. Data derived from HIV-1 transduced cells shows little alteration in the number of pseudogenes compared with mock transduced control; however, the number of differentially regulated pseudogenes was significantly higher in HIV-2-transduced cells compared with HIV-1. A total of 77 pseudogenes were differently regulated at 8 h and 93 at 12 h post-transduction; meanwhile, 78 pseuodogenic mRNA products were detected at 26 h. Amongst the HIV-2 altered pseudogenes, eukaryotic translation elongation factor alpha 1 (eEF1a1), protein disulfide isomerase family A member 3 (PDIA3) and nucleophosmin 1 (NPM1)-derived pseudogenic mRNA were detected. eEF1a1 has been described as cofactors for reverse transcription required for the stability of the reverse transcriptase complex [64,65]. During gp120 and CD4 interaction, disulfide isomerases initiate disulfide bond rearrangement in gp120, which aids the viral surface glycoprotein in its fusion with the cell membrane [66,67]. NPM1 is known to interact with HIV-1 Tat, and infection with HIV-1 induces acetylation of NPM1. This process is critical for the nuclear localization of Tat and for Tat-mediated transcription [68]. Pseudogenes were described as negative regulators of gene expression; it is plausible that the observed differential regulation of the expression of multiple pseudogenes by HIV-2 might contribute to its distinct pathophysiology.

GO analysis revealed further differences between HIV-1 and HIV-2-transduced cells. At 8 h, HIV-1 had a greater effect on cellular metabolism according to the significant GO terms observed, which included the positive regulation of metabolic processes, regulation of primary metabolic processes and positive regulation of RNA metabolic processes, among others. Meanwhile, HIV-2 mostly affects genes responsible for cell adhesion, extracellular matrix organization or keratinocyte differentiation. Increased metabolic activity promotes HIV-1 viral infectivity and replication, and previous studies have shown that inhibition of glycolysis would result in the inhibition of reverse transcription [69,70]. Rather surprisingly, we did not detect any significant GO term at 12 h post-infection in HIV-1-transduced cells. The number of differentially expressed genes was the lowest at 12 h compared with the 8 and 26 h time-points in HIV-1-infected HEK-293T cells. At the same time-point, HIV-2 managed to affect the genes involved in cell differentiation, animal organ development and system development. At 26 h post-transduction, HIV-1 managed to upregulate the expression of genes that are part of the response to oxidative stress, cellular response to toxic substances and detoxification. Compared with HIV-1, HIV-2 did not impact the cellular environment in an impactful way, only affecting genes that are related to the negative regulation of cell motility, negative regulation of cell migration and skin development.

Following the transcriptomic analysis, we utilized mass spectrometry to gain more information about changes in the cellular proteome generated by HIV in transduced cells. We found that the majority of the differentially regulated proteins were related to chaperon function, proteasome and mRNA processing. Our data showed that both HIV-1 and HIV-2 regulated the expression of 17 and 117 different proteins at 12 and 26 h post-transduction, respectively.

At 12 h, Zyx, PA2G4, hnRNPK and HSPA1A, among others, were significantly downregulated by both HIV-1 and HIV-2. Zyxin plays an important role in focal adhesion and actin polymerization; additionally, it was also found to be involved in many intracellular signaling pathways [71]. PA2G4, on the other hand, is a DNA/RNA binding protein that is involved in diverse cellular functions, most notably cell growth, apoptosis and differentiation. Previous studies have shown that this protein is downregulated by HIV-1 Vpr, potentiating G2 arrest and apoptosis in U87MG cells [72].

The hnRNP family of proteins mediates diverse functions in the cells, mostly involved in mRNA transcription, splicing, export, stability and translation. hnRNPQ, which was downregulated only by HIV-1, was shown to be involved in the proviral transcription of HIV-1 through interactions with viral protein Rev. It was also found to play an important role in RNA replication, splicing and mRNA stability of Hepatitis C virus [73].

Splicing factor SRSF2 and TRIM28 were also differently regulated by both pseudovirions at 26 h. SRSF2 was implicated in the downregulation of late steps of HIV-1 replication by mediating the splicing of tat and rev genes [74]. TRIM28, on the other hand, was shown to promote the latency of HIV-1 through the inhibition of positive transcription elongation factor (P-TEFb) and SUMOylating CDK9 along with SUMO4 [75]. hnRNPU, hnRNPK, hnRNPA2/B1, hnRNPQ, together with HSPA1A, DDX3X, SRSF1, NCL and TRIM28, were observed in complex with HIV mRNA, and with viral proteins in staufen1 ribonuclear complexes [76].

In contrast to HIV-1, HIV-2 downregulated NCAPH and PRRC2A, two proteins that are known to play an important role in cell proliferation and migration [77,78]. Retroviruses and, indeed, DNA and RNA viruses, in general, are known to modulate the cell cycle and cellular proliferation in order to facilitate viral replication and survival [79]. However, why only HIV-2 influenced these two key players at the 12 h time-point deserves further exploration. At the 26 h time-point, HIV-2 differentially regulated a slightly higher number of proteins compared with HIV-1. In comparison to HIV-1, HIV-2 upregulated members of the HSP70 (1A, 5 and 9), CCT3 and TFRC, to name a few. As a chaperone protein, HSP70 functions include mediating folding, translocation and degradation of target protein. In the context of HIV infection, it was shown that HSP70 exerts a dose-dependent inhibition on HIV-1 in CD4+ T cells [80]. Sugiyama et al. also reported that HSP70 inhibits the ubiquitination and degradation of APOBEC3G by Vif [81]. Moreover, by stimulating the binding of the HIV-1 matrix to karyopherin alpha, HSP70 is implicated in the nuclear import of PIC of HIV-1 [82]. CCT3, on the other hand, was shown to assist Gag polyprotein, enhancing the infectivity of Mason–Pfizer monkey virus (M-PMV) [83,84].

TFRC is a membrane glycoprotein that mediates the uptake of the transferrin–iron complex through receptor-mediated endocytosis [85]. It was shown that HIV infection leads to increased cellular iron levels, which enhances HIV replication [86]. HIV-1 Nef was shown to mediate the reduction of TFRC’s recycling rate, leading to its accumulation in early endosomes and, therefore, reduced expression at the cell surface in a T cell line [87]. However, this finding was not corroborated in another study, wherein Nef proteins of HIV-1 and its simian precursors did not affect the level of surface TFRC. However, Nef proteins of other lentiviruses reduced the internalization of the receptor in the myeloid and T cell line [88]. The differential regulation of TFRC might be dependent on the phase of the viral life-cycle and the type of the infected cell. The upregulation of TFRC by HIV-2 is indeed an interesting finding. Recently, TFRC was identified as a candidate entry protein for the Influenza A virus [89], and given the wide spectrum of receptor utilization by HIV-2, it may as well be utilized by the virus, although further studies are needed to confirm this hypothesis.

Compared to HIV-1, HIV-2 managed to downregulate the expression of RANBP1, which deserves significant attention. This protein was found to contain a nuclear export signal (NES) domain that is similar in structure and function to that found in HIV Rev; therefore, it is hypothesized that it shares a step in the post-transcriptional pathway with Rev [90]. Its downregulation by HIV-2 raises questions about whether it could be implicated in the suppression of the transport of proviral transcripts.

To further outline the differences between the two pseudovirions, the GO term analysis was carried out on significantly altered proteins in transduced cells. Very few GO terms were apparent at 12 h in the case of both viruses. At 26 h, however, there were several GO terms shared between HIV-1 and HIV-2, such as regulation of alternative mRNA splicing via the spliceosome, regulation of DNA-templated transcription elongation and post-transcriptional regulation of gene expression (Appendix A). While there were many shared terms between the two, many differences were also observed at that time-point. GO terms, such as cellular response to unfolded protein, regulation of gene expression and positive regulation of apoptotic processes, were detected in the case of HIV-1, while negative regulation of DNA metabolic processes, negative regulation of protein ubiquitination and positive regulation of translation were characteristic of HIV-2.

It is important to note that some proteins identified in the proteomic analysis did not correlate with the findings from the transcriptomic data. Transcriptomic changes are not always reflected at the level of proteins, given the substantial role of post-transcriptional and post-translational regulation governing protein production; also, given the methodology applied here and the relatively long intervals between the time-points [91], a correlation between transcriptome and proteome at the same time-point was unlikely. However, we could follow the regulation trends of many proteins across the time-points in the case of both pseudovirions, and many significant differences were observed (Appendix A).

In conclusion, the analysis of the proteomic and transcriptomic data from the early phase of HIV-1 and HIV-2-based pseudovirion-transduced HEK-293T cells indicates that the cellular response varies significantly between the two. Our analysis was focused on changes in HEK-293T cells, not primary immune cells, but we hope that the data gathered can be of aid in understanding the pathomechanistic aspect of transduction with lentivectors. Moreover, to our knowledge, studies of the effects of HIV transduction in the early phase of infection are indeed lacking, especially in the case of HIV-2. We hope that our study may provide insight into the understudied replication cycle of HIV-2 and enrich our knowledge about the use of HIV-based lentiviral vectors as a whole.

## 4. Materials and Methods

### 4.1. Plasmids and Vectors

A second-generation lentiviral vector system was used for HIV-1, HIV-2 and “mock” pseudovirion production. For production of HIV-1 pseudovirions, the following plasmids were used: pWOX-CMV-GFP transfer vector, which was modified to code for mCherry instead of green fluorescent protein (GFP), psPAX2 as a packaging plasmid (a kindly gift from Dr. D. Trono at the University of Geneva Medical School), and pMD.G encoding for the envelope protein of vesicular stomatitis virus. For the production of HIV-2 pseudovirions, the following plasmids were used: CGP, a ROD-based HIV-2 protein expression vector, CRU5SINCGW transfer vector, which has a GFP expression cassette under CMV promoter (both are a kindly gift from Joseph P. Dougherty at the Robert Wood Johnson Medical School) and pMD.G plasmid. For mock pseudovirion production, pTY-EFeGFP vector, a lentiviral transducing vector containing a GFP expression cassette under an EF1α promoter, and pMD.G vectors were used.

### 4.2. Production of HIV-1, HIV-2 and “Mock” Pseudovirion Particles

Human embryonic kidney (HEK-293T) cell line was obtained from American Type Culture Collection (ATCC) (Manassas, VA, USA). HEK-293T was maintained in Dulbecco’s modified Eagle’s medium (DMEM) (Sigma-Aldrich, St. Louis, MO, USA) containing 10% FBS, 1% L-glutamine and 1% penicillin/streptomycin.

For the production of HIV-1, HEK-293T cells were transfected with pWOX-CMV-mCherry, psPAX_2_ and pMD.G plasmids in a 3:2:1 ratio; for HIV-2 production CGP, CRU5SINCGW and pMD.G plasmids were used in a 1:1:1 ratio; for “mock” production, pTY-EFeGFP and p.MDG were used in a 1:1 ratio.

The day before transfection, HEK-293T was passaged in order to achieve around 70% confluence on the next day (~3 × 10^6^ cell/flask). Transfection was carried out using the polyethylenimine (PEI) method. Following transfection, cells were incubated at 37 °C, 5% CO_2_ in antibiotic-free DMEM media containing 1% FBS. After 5–6 h of incubation, the media was replaced with DMEM containing 10% FBS, 1% penicillin–streptomycin and 1% glutamine. The supernatant containing virions was collected and filtered through a 0.45-μm polyvinylidene fluoride filter (Merck Millipore, Darmstadt, Germany) after 24, 48 and 72 h; then, the supernatants were pooled together and concentrated by ultracentrifugation (100,000× *g* for 2 h 10 min at 4 °C), and the viral pellet was dissolved in ~ 300 µL PBS and stored at −70 °C. Reverse transcriptase activity was measured using an enzyme-linked immunosorbent assay (ELISA)-based colorimetric assay (Roche Applied Science, Mannheim, Germany). Quantification of virions was carried out by transduction experiments on HEK-293T cells using the collected virions and measuring the transduction units/mL (TU/mL). The amount was then adjusted to RT equivalence of HIV-1 pseudovirions in order to use an equal quantity of virions for transduction experiments.

### 4.3. Transduction of HEK-293T Cells for Transcriptomic Analysis

The day before transduction, HEK-293T cells were plated in 6 well plates (5 × 10^5^ cells/well) in DMEM supplemented with 10% FBS, 1% L-glutamine and 1% penicillin–streptomycin. The following day, the medium was discarded, and the cells were transduced with 5 ng RT-equivalent of HIV-1/HIV-2 or ”mock” pseudovirions in serum and antibiotic-free media complemented with 8 µg/mL polybrene. Cells were collected at 8, 12 and 26 h after transduction; media was discarded, and cells were washed with PBS and then suspended into TRIzol reagent (Thermo Fisher Scientific, Waltham, MA, USA). RNA isolation was carried out according to the manufacturer’s instructions. The quality of the RNA was determined by Agilent RNA 6000 Nano kit on an Agilent 2100 Bioanalyzer (Agilent Technologies, Waldbronn, Germany). Thereafter, high-throughput sequencing was performed on the MGI DNBSEQ G400 sequencer using MGIEasy RNA Library Prep Set at the Genomic Medicine and Bioinformatics Core Facility of the University of Debrecen.

### 4.4. Transduction of HEK-293T Cells for Proteomic Analysis

A day before transduction, HEK-293T cells were seeded in T-25 flask (0.7–1 × 10^6^ cells/flask) in 5 mL of DMEM supplemented with 10% FBS, 1% L-glutamine and 1% penicillin–streptomycin. The next day, the medium was discarded, and the cells were transduced with 15 ng RT-equivalent of HIV-1/HIV-2 or “mock” pseudovirions in serum and antibiotic-free media complemented with 8 µg/mL polybrene. Cells were thereafter incubated at 37 °C, 5 % CO_2_ for eight, 12 and 26 h. Next, the media was discarded, and cells were mechanically detached and suspended in PBS. After brief centrifugation, the pellet was stored at −20 °C until further analysis.

### 4.5. Transcriptomic Data Analysis

Technical description of data and sequencing statistics is provided in Appendix A. RNA-seq raw fastq data were cut from adaptors and quality trimmed (phred score 30) using Trimmomatic v0.36 [92]. The minimal trimmed read length was set to 36 bp. Quality check was carried out with FastQC v0.11.9 [93]. The reads were mapped to the GRCh38 Human Genome Assembly reference genome using Hisat2 v2.1.0 [94]. Aligned reads were counted using FeatureCounts v2.0.1 [95]. The obtained count matrix was used for analysis with R v4.2.3 [96]. Counts normalized with the median-of-ratios method were used for estimating differential expression (DE) with DESeq2 v1.38.3 [97]. To improve the accuracy of fold change estimates, we applied the “Adaptive Shrinkage” package version 2.2-63 [98]. The downstream analysis consisted of genes passing significance filters with adjusted *p*-value (padj) < 0.05 and absolute log_2_ fold-change (abs(LFC)) > 1. Gene ontology (GO) enrichment analysis of the DE genes was performed with clusterProfiler package [99].

### 4.6. GeLC-MS/MS Analyis

Lysis of the HIV-1, HIV-2 or mock transduced cells was performed in 100 µL lysis buffer (50 mM Tris pH 8.3, 1 mM EDTA, 17 mM β-mercaptoethanol, 0.5% (*v*/*v*) Triton-X100) using three freeze–thaw cycles. Bradford method was used to determine the protein concentration, and 100 µg of protein in each case was subjected to in-gel digestion followed by liquid chromatography–tandem mass spectrometry (GeLC–MS/MS) analysis [100]. Briefly, samples were run into a 5% SDS-polyacrylamide gel using a 100 V current for 20 min. The proteins were stained with PageBlue Protein Staining solution (Thermo Scientific, Waltham, MA, USA), and the stained gel slice was excised, separated into three equal portions and subjected to in-gel trypsin digestion. Reduction was performed with 20 mM dithiothreitol (Bio-Rad, Hercules, CA, USA) for 1 h at 56 °C, followed by alkylation with 55 mM iodoacetamide (Bio-Rad, Hercules, CA, USA) for 45 min at room temperature in the dark. Overnight trypsin digestion was performed at 37 °C using stabilized MS-grade TPCK-treated bovine trypsin (ABSciex, Framingham, MA, USA). The digested peptides were extracted and dried in a speed-vac (Thermo Scientific, Waltham, MA, USA). The peptides were re-dissolved in 33 μL 1% formic acid (VWR Ltd., Radnor, PA, USA) before LC–MS/MS analysis. The peptide concentration of the samples was determined using the BCA method. Prior to mass spectrometry analyses, the samples were spiked with equal amounts of indexed retention time (iRT) peptide mixtures (Biognosys, Schlieren, Switzerland), and the samples were analyzed in duplicate.

Prior to the mass spectrometric analysis, peptides were separated in a 180 min water/acetonitrile gradient using an Easy nLC 1200 nano UPLC (Thermo Scientific, Waltham, MA, USA). The peptide mixtures were desalted in an ACQUITY UPLC Symmetry C18 trap column (20 mm × 180 µm, 5 μm particle size, 100 Å pore size, Waters, Milford, MA, USA), followed by separation in Acclaim PepMap RSLC C18 analytical columns (150 mm × 50 μm × 2 μm particle size, 100 Å pore size, Thermo Scientific, Waltham, MA, USA). Chromatographic separation was performed using a gradient of 5–7% solvent B over 5 min, followed by a rise to 15% of solvent B over 50 min and then to 35% solvent B over 60 min. Thereafter, solvent B was increased to 40% over 28 min and then to 85% over 5 min, followed by a 10 min rise to 85% of solvent B, after which the system returned to 5% solvent B in 1 min for a 16 min hold-on. Solvent A was 0.1% formic acid in LC water (Sigma, St. Louis, MO, USA); solvent B was 95% acetonitrile (Sigma, St. Louis, MO, USA) containing 0.1% formic acid. The flow rate was set to 300 nL/min.

Data-dependent analyses were carried out on an Orbitrap Fusion mass spectrometer (Thermo Scientific, Waltham, MA, USA). The 14 most abundant multiply charged positive ions were selected from each survey MS scan using a scan range of 350–1600 *m*/*z* for MS/MS analyses (Orbitrap analyzer resolution: 60,000, AGC target: 4.0 × 10^5^, acquired in profile mode). Collision-induced dissociation (CID) fragmentation was performed in the linear ion trap with 35% normalized collision energy (AGC target: 2.0 × 10^3^, acquired in centroid mode). Dynamic exclusion was enabled during the cycles (exclusion time: 45 s).

### 4.7. Data analysis of Mass Spectrometry

The acquired LC-MS/MS data were used for protein identification with the help of MaxQuant 1.6.2.10 software [101] searching against the Human SwissProt database (release: 2020.02, 20,394 sequence entries), the HIV-1 and HIV-2 SwissProt databases (release: 2020.02, 381 sequence entries for HIV-1 and 109 sequence entries for HIV-2), and against the contaminants database provided by the MaxQuant software ver. 1.6.2.10. Cys carbamidomethylation, Met oxidation and N-terminal acetylation were set as variable modifications. A maximum of 2 missed cleavage sites were allowed. Results were imported into Scaffold 4.8.9 software (ProteomeSoftware Inc., Portland, OR, USA). Proteins were accepted with at least 3 identified peptides using 1% protein false discovery rate (FDR) and 0.1% peptide FDR. For label-free quantification, the normalized total precursor intensities were used, and quantitative values of the identified proteins were normalized to the quantitative values of the iRT mixture.

### 4.8. Proteomic Data Analysis

The list of identified proteins with intensity values was exported from Scaffold and further processed in the R environment (v4.3.1) [96]. Normalized values were calculated accounting for variations in overall sample protein concentrations (determined post-digestion by the BCA method), injection volumes and the sum of detected protein intensities per sample. We employed a series of mixed-effects ANOVA models, one for each protein, to identify the statistically significant differences in protein abundances between the investigated groups. Sample and measurement replicates were treated as random effects, while the different transduced cell groups were modeled as fixed effects [102]. We ran the computations utilizing the emmeans (v1.8.8) and lme4 (1.1.34) packages [103]. After the linear model fitting, post-hoc tests were applied to determine the *p*-values of group differences and significant results with an FDR < 0.05 criteria were retained. The protein coding gene annotation was retrieved from the daily updated ‘gene_info’, ‘gene2go’ files available on the NCBI FTP site (https://ftp.ncbi.nlm.nih.gov/gene/DATA/, accessed on 15 November 2023) and reorganized by in-house shell scripts. We carried out enrichment analysis for gene ontology (GO) with the topGO R package (v2.52.0) using a custom-generated gene-to-GOs mapping file [104]. The protein universe consisted of the entire set of proteins (807) detected in all 26 h post-transduction samples; significant proteins were filtered according to the FDR < 0.05 threshold. In the topGO algorithm, we chose the default ‘weight01’ method with the Kolmogorov–Smirnov statistical test and selected the enriched GO terms having FDR-adjusted *p*-values < 0.05. Data visualization was performed with ggplot2 (v3.4.3), ggrepel (v0.9.3), ggpubr (v0.6.0), GOplot (v1.0.2) [105], ggnewscale (v0.4.9), VennDiagram (v1.7.3) R packages.

## Figures and Tables

**Figure 1 ijms-25-00380-f001:**
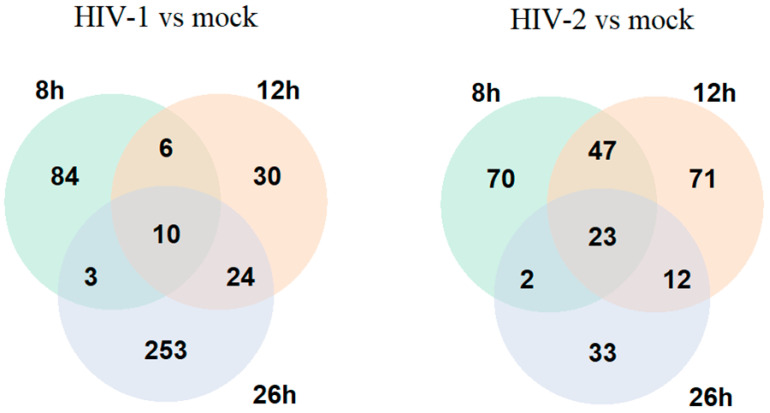
Venn diagram showing the differential gene expression of protein-coding genes between HIV-1 and mock or HIV-2 and mock-transduced samples in all of the observed time points. log2 fold change (LFC) > 1 adj. *p*-value < 0.05.

**Figure 2 ijms-25-00380-f002:**
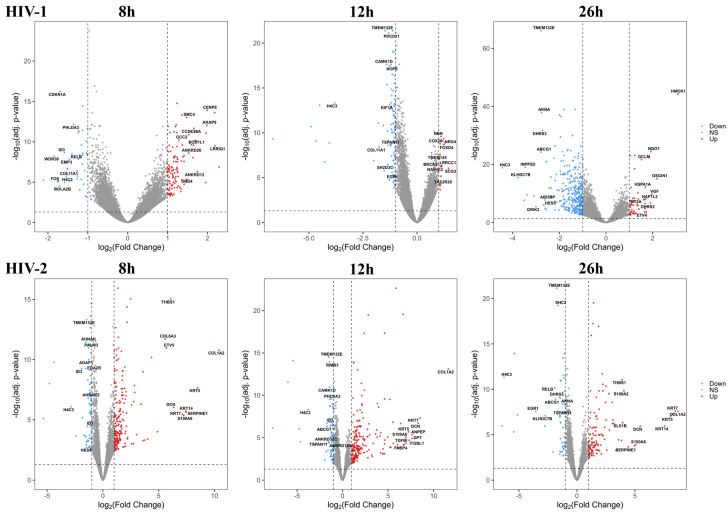
Volcano plot showing the altered genes from HIV-1 (**top**) and HIV-2 (**bottom**) transduced HEK293T cells compared with the mock control throughout the different time-points. The log2 fold change (log2FC) of the detected genes is shown on the x-axis, while the log10 adjusted *p*-value of each gene is present on the y-axis. Number of dots represent the detected genes from each sample, while the color corresponds to the significantly up (red) or downregulated (blue) genes. Grey dots represent the non-significant transcripts.

**Figure 3 ijms-25-00380-f003:**
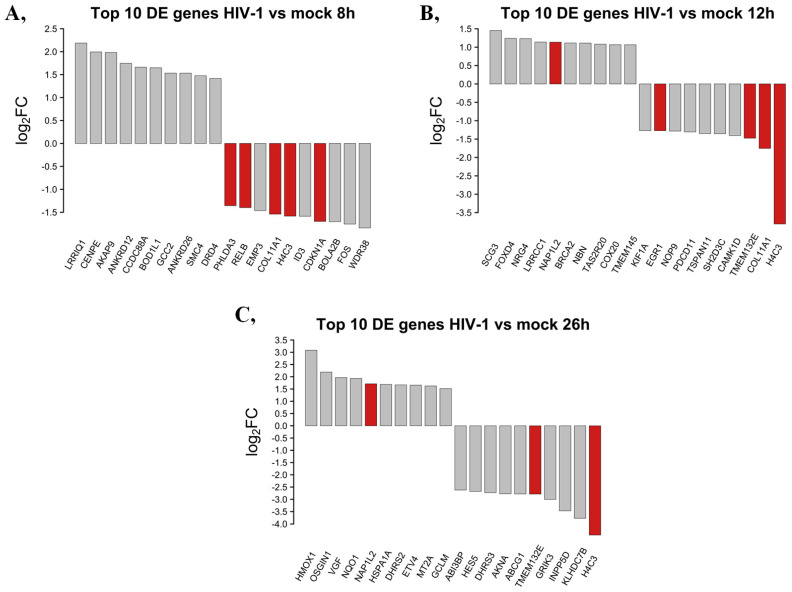
Top 10 up- and downregulated DEGs at the 8 (**A**), 12 (**B**) and 26 (**C**) hour time-points following transduction with HIV-1. log2FC of the up- and downregulated transcripts coding for proteins in HIV-1 transduced HEK-293T cells compared to mock transduction. Red bars show genes that are regulated similarly in all observed time-points compared with mock-transduced samples, while grey bars correspond to the significantly altered transcripts that were regulated only in the given time-point.

**Figure 4 ijms-25-00380-f004:**
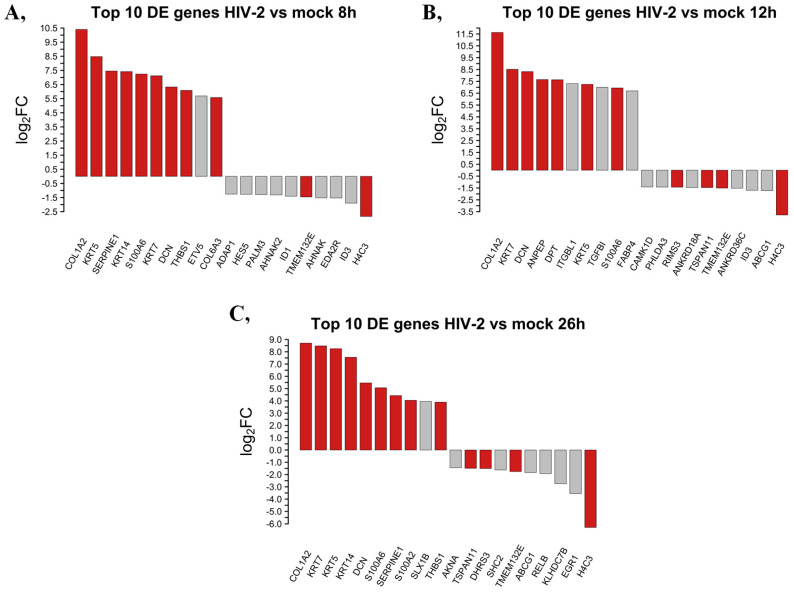
Top 10 up- and downregulated DEGs at the 8 (**A**), 12 (**B**) and 26 (**C**) hours time-points following transduction with HIV-2. log2FC of the up- and downregulated transcripts coding for proteins in HIV-2 transduced HEK-293T cells, compared to mock transduction. Red bars show genes that are regulated similarly in all observed time points, while grey bars correspond to the significantly altered transcripts that were regulated only in the given time-point.

**Figure 5 ijms-25-00380-f005:**
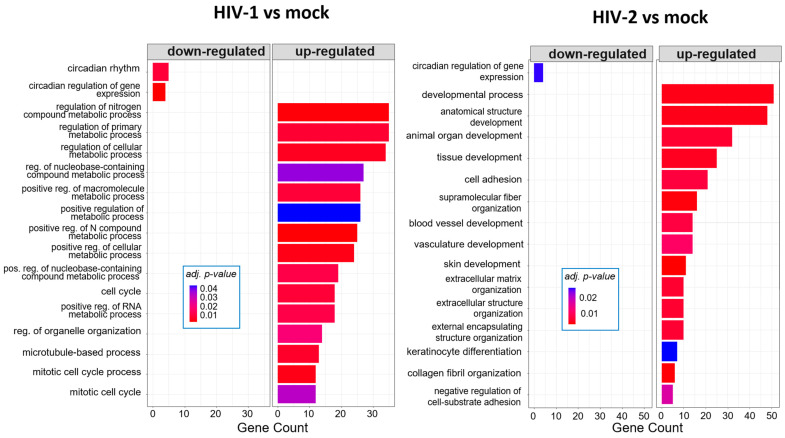
Gene ontology (GO) analysis of DEGs in HIV-1 (**left**) and HIV-2 (**right**) transduced cells at 8 h time-point. Top genes classified according to significant enrichment terms. Color intensity corresponds to the significance of each term. Counts represent the number of differentially expressed genes associated with the listed gene ontology ID.

**Figure 6 ijms-25-00380-f006:**
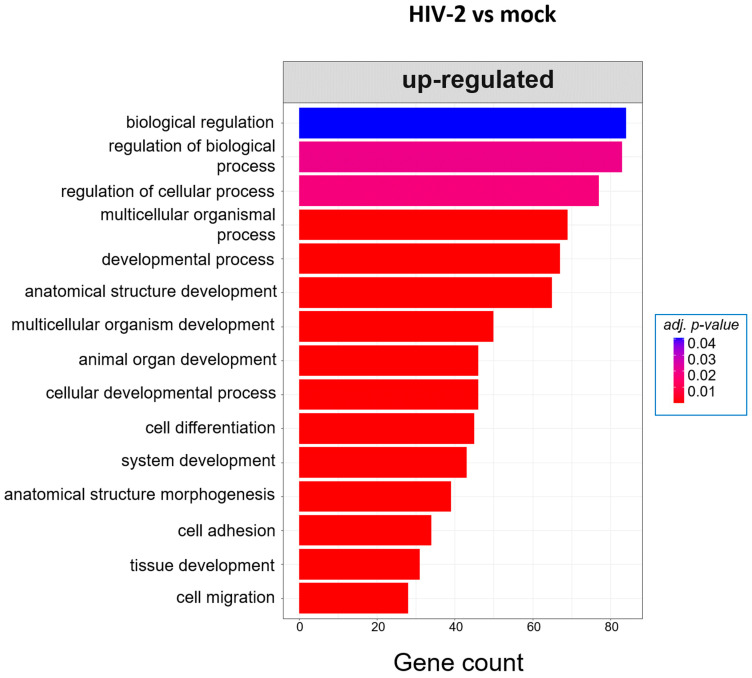
Gene ontology (GO) analysis of DEGs in HIV-2-transduced cells at 12 h time-point. Top genes classified according to significant enrichment terms. Color intensity corresponds to the significance of each term. Counts represent the number of differentially expressed genes associated with the listed gene ontology ID.

**Figure 7 ijms-25-00380-f007:**
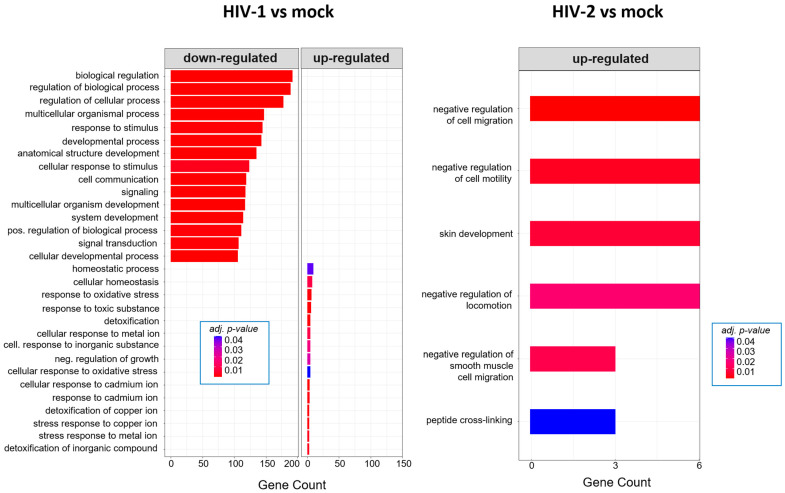
Gene ontology (GO) analysis of DEGs in HIV-1 (**left**) and HIV-2 (**right**) transduced cells at the 26 h time-point. Top genes classified according to significant enrichment terms. Color intensity corresponds to the significance of each term. Counts represent the number of differentially expressed genes associated with the listed gene ontology ID.

**Figure 8 ijms-25-00380-f008:**
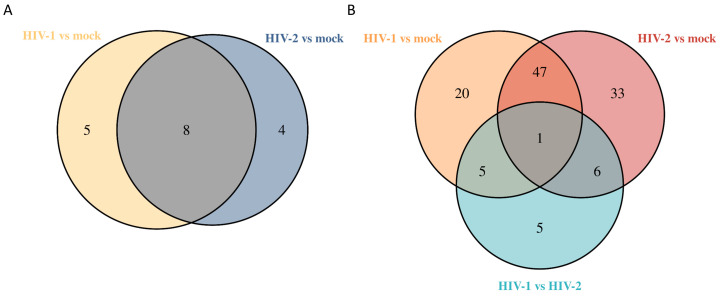
Venn diagram showing differential protein expression between HIV-1 vs. mock or HIV-2 vs. mock transduced samples at 12 (**A**) and additionally between HIV-1 vs. HIV-2 at 26 (**B**) hours post-transduction.

**Figure 9 ijms-25-00380-f009:**
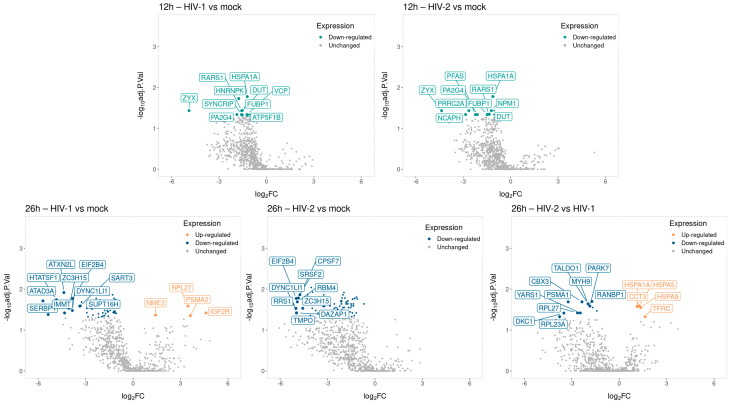
Volcano plot illustrating the changes in protein abundance in HIV-1 and HIV-2 transduced cells at 12 and 26 h post-transduction. The x-axis displays the log2 fold change (log2FC) of detected proteins, while the y-axis shows the log10 adjusted *p*-value for each protein. Each dot represents a detected protein, with color indicating significant upregulation (light orange) or downregulation (green, blue), and grey dots representing non-significantly regulated proteins. Note that only the top ten significantly regulated proteins, either up or downregulated, are labeled in the plot for each category.

**Figure 10 ijms-25-00380-f010:**
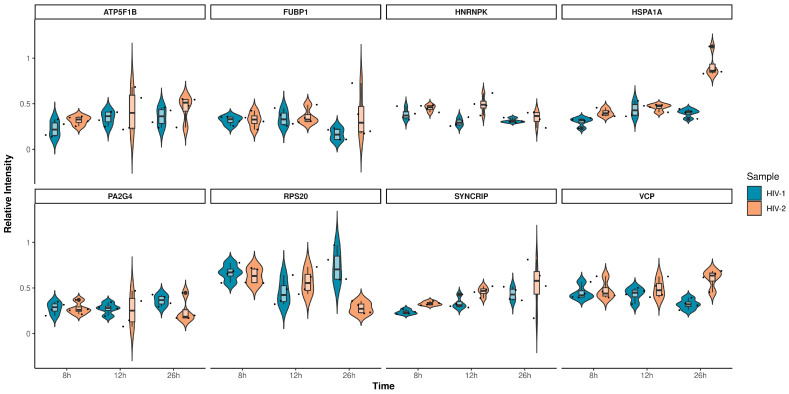
Violin plots depicting the intensity distribution of differentially regulated proteins across all observed time-points for HIV-1 (blue) and HIV-2 (light orange) transduced cells. The x-axis represents the distinct time-points, while the y-axis corresponds to the protein’s intensity.

**Figure 11 ijms-25-00380-f011:**
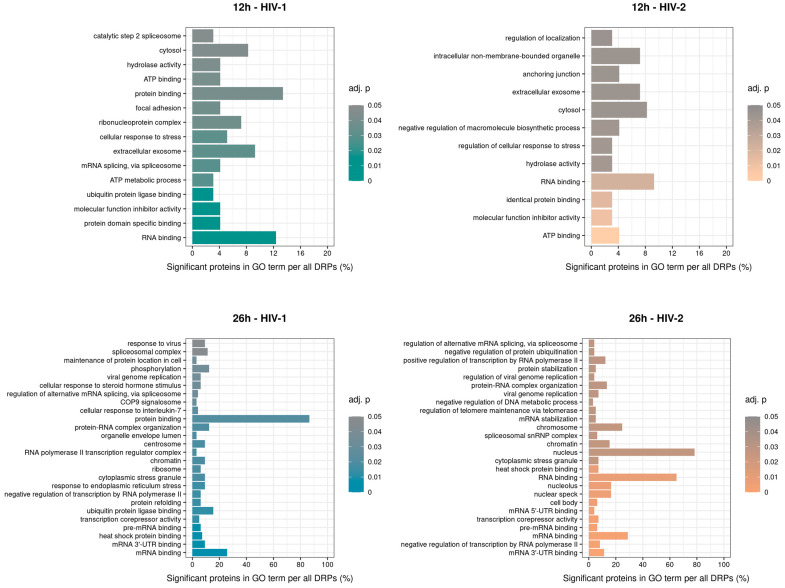
Classification of gene ontology (GO) terms of the enriched significant proteins from HIV-1 (**left**) and 2 (**right**) infected cells 12 (**top**) and 26 (**bottom**) hours post-transduction compared with the mock. Color intensity refers to the adjusted *p*-value of each term, while the x-axis is the percentages of significant proteins in GO term per all differentially expressed proteins (DEPs). The names of the enriched GO terms are on the y-axis.

**Figure 12 ijms-25-00380-f012:**
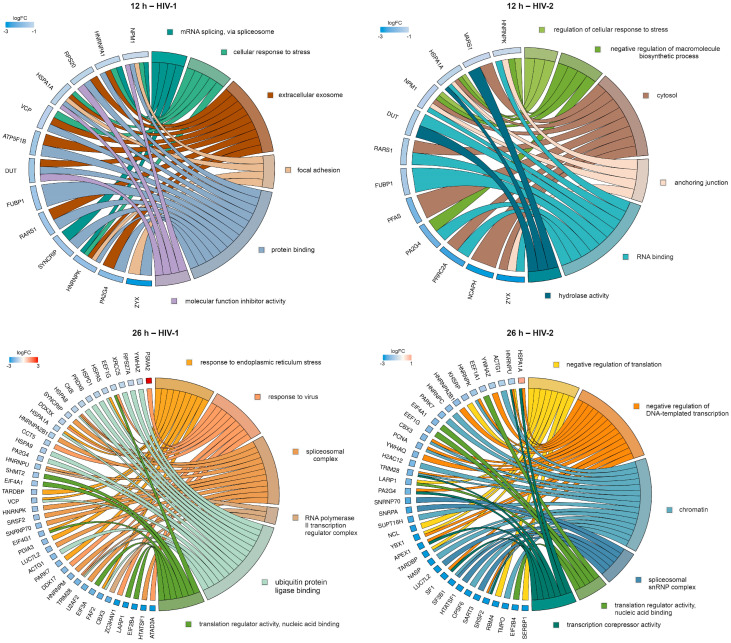
Protein association analysis. This figure illustrates the association between six selected GO terms and differentially regulated proteins (DRPs) by HIV-1 (**left**) and HIV-2 (**right**) at 12 (**top**) and 26 (**bottom**) hours post-transduction. The chosen GO terms encompassed mRNA splicing via spliceosome, cellular response to stress, extracellular exosome, focal adhesion, protein binding and molecular function inhibitor activity for HIV-1. In regard to HIV-2, at 12 h post-transduction, the selected terms included regulation of cellular response to stress, negative regulation of macromolecule biosynthetic process, cytosol, anchoring junction, RNA binding and hydrolase activity. At 26 h post-transduction, the selected terms were response to endoplasmic reticulum stress, response to virus, spliceosomal complex, RNA polymerase II transcription regulator complex, ubiquitin protein ligase binding, translation regulator activity, nucleic acid binding for HIV-1. In the case of HIV-2, the terms comprised negative regulation of translation, negative regulation of DNA-templated transcription, chromatin, spliceosomal snRNP complex, translation regulator activity, nucleic acid binding and transcription corepressor activity. The color intensities of the rectangles reflect the logarithm of the fold change values, representing the extent of changes in corresponding protein levels relative to the control. Notably, at 26 h post-transduction, the majority of DRPs are observed in the HIV-2 vs. Mock comparison. However, some DRPs were selectively identified from the HIV-2 vs. HIV-1 comparison, exemplified by instances such as HSPA1A being upregulated compared with HIV-1 (shown here) but downregulated compared with mock.

## Data Availability

Proteomic data presented in this study are openly available in MassIVE at ftp://massive.ucsd.edu/MSV000092581/, reference number PXD044317 accessed on 3 August 2023. Transcriptomic data presented in this study are openly available in GEO at https://www.ncbi.nlm.nih.gov/geo/query/acc.cgi?acc=GSE246164, reference number GSE246164 accessed on 25 October 2023.

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
