# Peer review of "Comparative Analysis of Differential Cellular Transcriptome and Proteome Regulation by HIV-1 and HIV-2 Pseudovirions in the Early Phase of Infection"

_ijms, 2023, doi:10.3390/ijms25010380_

Round 1

Reviewer 1 Report

Comments and Suggestions for Authors

Please see attached for detailed suggestions and comments.

Comments on the Quality of English Language

English writing is good in general.

Author Response

Dear Reviewer, Thank you for your effort and time spent revieweing our manuscript. Please find the attached response.

Reviewer 2 Report

Comments and Suggestions for Authors

In this paper, the authors transduced HEK-293T cells using HIV-1, HIV-2 and GFP mock pseudo virions and then they do RNA-Seq to check the up and down regulated genes after 8-, 12-, and 26-hours transduction. They found there was a difference between the HIV-1 and HIV-2 pseudo virions. They also investigated the difference of proteomics between HIV-1 and HIV-2 transduction in HEK 293T cells. The results showed that HIV-2 significantly downregulated the expression of certain proteins that perhaps benefited its life-cycle and latency, while HIV-1 influenced the cellular level of oxidative stress and proteins involved in cell death. The special comments were listed:

1.     It seems the up/down regulated genes in RNA do not match the protein changes. So, what makes the protein expression level change?

2.      There is a paper doing exactly the same thing, just early time point, like 2 hours after transducing. So, what is the meaning of doing late time point? Actually, investigating HIV in HEK293T cells, not in primary cells, is not really helpful to understand the mechanisms.

Author Response

Dear reviewer. Thank you indeed for your time and comments regarding our manuscript. Please find attached response to your comments.

Reviewer 3 Report

Comments and Suggestions for Authors

Proteo-transcriptomic analyses were employed in this study to determine the differential gene expression and proteomic changes induced by transduction with HIV-1/2 pseudovirions at 8, 12, and 26 hour time points in HEK-293T cells. For a  further revision, the following points should be considered.

1. It is important to select key genes and conduct experimental validation, e.g. qRT-PCR, western blot.

2. The RNA was extracted for transcriptome sequencing, which is inconsistent with the description of “genome” in the title and abstract.

3. The functional enrichment analysis can be enhanced by selecting the differentially expressed genes that are shared between the transcriptome and proteome.

SPECIFIC COMMENTS:

L52 Could you provide a more detailed explanation of the term 'RNA genome'.

L106-8 Please rewrite this sentence to clarify.

What’s the meaning of LFC in the caption of Figure 1?

L251-2 Meanwhile, HIV-2 differentially regulated the cellular expression of genes which are involved in the cell adhesion, extracellular matrix organization and tissue development. is confused.

The words are too small to pick out in Figures 5-7.

L292  a redundant word however.

Author Response

Dear Reviewer. Thank you for your input regarding our manuscript. Please find attached response. We hope that you may find it satisfactory.

Round 2

Reviewer 1 Report

Comments and Suggestions for Authors

All my questions are addressed.

Comments on the Quality of English Language

English writing is good in general.

Author Response

Dear Reviewer,

We would like to thank you again for your time and valuable comments that have vastly improved our manuscript.

Reviewer 2 Report

Comments and Suggestions for Authors

Thank you for the opportunity to review this revised manuscript. The authors have revised the manuscript in accordance with the comments. Still have one comment.

Do authors analyze 8-hour transcriptomic data with 26-hour proteome since the author said fewer overlapping genes is due to the delay?

Author Response

Dear Reviewer, attached please find our response. Thank you.
